# Growth Performance and Biochemical Profiles of Fairy Shrimp (*Streptocephalus sirindhornae*) Fed Natural Diets at Low and High Stocking Densities

**DOI:** 10.3390/biology15020117

**Published:** 2026-01-07

**Authors:** Kosit Sriphuthorn, Naiyana Senasri, Prapatsorn Dabseepai

**Affiliations:** 1Department of Fisheries, Faculty of Natural Resources, Rajamangala University of Technology Isan, Sakon Nakhon Campus, Sakon Nakhon 47160, Thailand; naiyana.se@rmuti.ac.th; 2Applied Taxonomic Research Center, Department of Biology, Faculty of Science, Khon Kaen University, Khon Kaen 40002, Thailand; prapda@kku.ac.th

**Keywords:** amino acids, Anostraca, carotenoids, fatty acids, gut content, live food, omega 6, phytoplankton, zooplankton

## Abstract

The fairy shrimp *Streptocephalus sirindhornae* is a small freshwater crustacean that can be used as natural live feed for fish and prawn farming. This study examined how rearing density and available food organisms affect its growth and nutritional quality. Lower densities reduced competition and water quality stress, allowing the shrimp to grow faster and reach larger sizes. Microscopic observation revealed that they mainly fed on green algae such as *Chlorella* sp. and *Monoraphidium* sp., which are rich in protein and pigments that promote growth. The fairy shrimp also contained high levels of essential amino acids, beneficial fatty acids, and carotenoids, indicating their high nutritional value. They selectively consumed algae species with higher nutrient content instead of all plankton available. These findings improve understanding of how fairy shrimp adapt their feeding behavior to their environment and provide useful insights for aquaculture management. Promoting the use of *S. sirindhornae* as live feed may support sustainable production of nutritious food for freshwater aquaculture species.

## 1. Introduction

The fairy shrimp (Order Anostraca), with approximately 300 recognized species globally [1], offers potential for several technical uses [2]. Some freshwater fairy shrimps are a potential nutritional food source for fish and crustaceans due to their high nutritive content [2,3]. In Thailand, three species of fairy shrimp have been identified in temporary freshwater habitats: *Streptocephalus sirindhornae*, *Branchinella thailandensis*, and *Streptocephalus siamensis* [4,5,6]. Nonetheless, only *S. sirindhornae* and *B. thailandensis* have been recognized as live feed for aquaculture in the nation [7,8]. Among these, *Streptocephalus sirindhornae* is nutritionally rich, containing protein, essential amino acids, and essential fatty acids, as well as carotenoids [9]. For this reason, it is used as an alternative feed source for aquaculture, where both nauplii and adult shrimp have been used as live feed for giant freshwater prawn, *Macrobrachium rosenbergii* [10,11]. Additionally, the dried adult *S. sirindhornae* has been used as a carotenoid supplement in pellet diets for freshwater prawn, *M. rosenbergii* [8], and flowerhorn cichlid (*Amphilophus citrinellus* × *Cichlasoma trimaculatum*) [12]. Studies on the growth and reproductive performance of crayfish (*Procambarus clarkii*) fed frozen adult *S. sirindhornae* have been conducted [13].

Efforts to enhance the biomass production of *S. sirindhornae* have focused on both artificial concrete ponds and earthen ponds [14]. However, the lack of appropriate methods of rearing the fairy shrimp remains a problem in the production process [1]. Several factors contribute to biomass production in anostracans, among which temperature is significant, as it affects the life cycle, growth, longevity, and reproduction of *Branchipus schaefferi* [15]. Food availability is another key factor for production performance in fairy shrimp. They can utilize several sources of organic effluents as food due to their filtration capacity [16]. Additionally, the effects of foods containing different levels of phosphorus on the survival and growth of *S. sirindhornae* have been reported [17]. The effects of stocking density on the biomass production of fairy shrimp cultured with fish effluent have also been reported [16,18,19]. However, there is conflicting evidence regarding different stocking densities for *S. sirindhornae* and *B. thailandensis* when cultured with fish effluent [18,19]. The information on the biochemical composition of *S. sirindhornae* fed natural food is limited.

Therefore, this study investigated the growth performance of *S. sirindhornae* cultured with natural foods in earthen ponds at low and high stocking densities. Additionally, food items in the guts of the fairy shrimp and the plankton composition in the earthen ponds were investigated. Furthermore, at the end of the experiments, the biochemical profiles of the cultured fairy shrimp were examined to evaluate their nutritional quality and potential use as live feed.

## 2. Materials and Methods

### 2.1. Pond Preparation and Natural Food Production

The earthen ponds were first sun-dried to eliminate unwanted organisms, and the bottom substrate was subsequently disinfected. Hydrated lime was applied evenly at 100 kg ha^−1^ (approximately 16 kg rai^−1^) to improve the pond bottom quality prior to culture initiation. The ponds were then filled with earthen water to a depth of 30–50 cm and filtered through a double-layered green cloth net (24–26 mesh inch^−1^) to prevent the entry of predators and unwanted larvae.

Fresh green algae (*Chlorella* sp.) were added at 10% of the total water volume to enhance natural food availability. *Chlorella* sp. was cultured outdoors under continuous aeration in a medium containing N–P–K (16–20–0; 150 g), urea (300 g), lime (CaCO_3_; 90 g), and rice bran (500 g) per m^3^ of water. After 3–5 days, the culture reached a density of approximately 1.2 × 10^6^ cells mL^−1^ and was subsequently used as inoculum. Additionally, dried chicken manure (70 kg) and N–P–K fertilizer (10 kg) were applied to the ponds to promote algal maturation following [8,20], typically requiring 5–7 days.

Floating cages (1 × 2 m) were constructed by installing four vertical posts into the pond bottom, with the upper ends extending 50 cm above the water surface. A fish net was suspended from a surrounding rope, ensuring a minimum submerged depth of 0.5 m. The cages were carefully lowered and secured with strong cords.

Fairy shrimp were cultured at two stocking densities, 20 and 40 individuals L^−1^, each with three replicates. Growth measurements (body length and weight) were conducted on days 0, 3, 6, 9, 12, and 15. The selected densities were based on published studies on branchiopods (10–50 ind. L^−1^) and preliminary observations indicating that densities above 40 ind. L^−1^ reduced swimming activity, slowed growth, and increased food competition, whereas densities below 20 ind. L^−1^ yielded insufficient biomass. Thus, 20 and 40 ind. L^−1^ were chosen to represent low and high densities for evaluating effects on growth, biochemical composition, and feeding patterns.

### 2.2. Fairy Shrimp Culture

Nauplii of *Streptocephalus sirindhornae* were obtained from the Applied Taxonomic Research Center, Khon Kaen University (Khon Kaen, Thailand). Eggs were hatched in plastic tanks (50 × 70 cm) filled to a depth of 15–20 cm and incubated for 24 h. Newly hatched larvae were siphoned and released into a 5 × 5 m earthen pond (water depth: 10 cm). Prior to larval release, *Chlorella* sp. was added at 50 L, followed by daily supplementation of 150 L (culture density: ~1.2 × 10^6^ cells mL^−1^) to maintain natural food availability. Before the experiment, fairy shrimp were pre-reared in concrete ponds for 15 days.

For the experiment, fairy shrimp were transferred into 1 m^3^ floating cages suspended in earthen ponds and stocked at two densities: 20 and 40 ind. L^−1^, with three replicates per treatment. Stocking densities were selected based on preliminary trials and published studies on anostracan culture. Throughout the 15-day experimental period, shrimp relied solely on naturally occurring pond plankton, with no artificial feed supplementation. Growth measurements (total length and wet weight) were recorded on days 0, 3, 6, 9, 12, and 15.

### 2.3. Plankton Sampling and Identification

Plankton samples were collected every three days throughout the 15-day experimental period from three fixed sampling points within each pond. At each point, 20 L of water were filtered through a 30 µm mesh plankton net, and the retained materials were immediately preserved in 4% formalin. Phytoplankton and zooplankton were identified under a compound microscope (Olympus CX23; Olympus Corporation, Tokyo, Japan) following standard taxonomic keys and identification guides [21,22,23,24,25,26].

### 2.4. Gut Content Composition Analysis

Gut content composition analysis followed the procedure of [27]. Every three days, ten fairy shrimp from each replicate (totaling 30 individuals per treatment) were randomly collected, and their alimentary canals were carefully dissected. Due to the small biomass of individual guts, the ten alimentary canals from each replicate were pooled into a single composite sample and mounted on a glass slide with a few drops of distilled water to facilitate separation of food particles. Phytoplankton and zooplankton present in the gut contents were identified following the same taxonomic procedures described in Section 2.3.

### 2.5. Growth Measurements

Ten fairy shrimp from each replicate were randomly selected on each sampling day (days 3, 6, 9, 12, and 15). Total length was measured from the anterior margin of the head to the distal end of the telson using a Vernier caliper (precision: 0.01 mm). Body weight was recorded for the same individuals using a digital balance (PL202-S, Mettler Toledo, Greifensee, Switzerland; readability: 0.01 g).

### 2.6. Water Quality Measurements

Water quality parameters were measured every three days. Temperature (°C), pH, conductivity (µS cm^−1^), and total dissolved solids (mg L^−1^) were measured using a multiparameter meter (HI98130, Hanna Instruments, Woonsocket, RI, USA). Dissolved oxygen (mg L^−1^) was measured with a YSI Pro20i meter (Xylem Inc., Rye Brook, NY, USA). Salinity (ppt) was assessed using an Atago refractometer (Cat. No. 2441). Turbidity (NTU) was determined with a TN-100 turbidity meter (Thermo Scientific/Eutech, Waltham, MA, USA). Nitrite and ammonia concentrations (mg L^−1^) were determined colorimetrically using a Hach DR/2400 spectrophotometer (Hach Company, Loveland, CO, USA) following diazotization and salicylate methods.

### 2.7. Amino Acid Analysis

Amino acid composition was determined following acid hydrolysis and ion-exchange chromatography as described by [28]. Samples underwent acid oxidation prior to hydrolysis, and amino acids were analyzed using GC–MS (Agilent 6890N GC; 5973 MS) equipped with a Phenomenex Zebron ZB-AAA column. Protein-bound tryptophan was not quantified due to degradation during acid hydrolysis. Because the biomass obtained from each replicate was insufficient, shrimp samples were pooled across stocking densities prior to analysis.

### 2.8. Fatty Acid Analysis

Fatty acid composition was determined using a modified hydrolytic method based on AOAC Official Method 996.06 [28]. Lipids were extracted using diethyl ether. Although chloroform–methanol mixtures are widely used for comprehensive lipid extraction, ether-based extraction protocols are also commonly applied in practical aquaculture studies and were considered suitable for the present samples. Extracted lipids were converted to fatty acid methyl esters (FAMEs) by methylation with boron trifluoride (BF_3_) in methanol.

FAMEs were analyzed by gas chromatography–mass spectrometry (GC–MS; Agilent 6890N GC coupled with a 5973 MS detector (Agilent Technologies, Inc., Santa Clara, CA, USA)) equipped with an SP-2560 capillary column (100 m × 0.25 mm ID × 0.20 µm film thickness). Helium was used as the carrier gas. Injection was performed in split mode at 260 °C. The oven temperature program was set at 140 °C for 5 min, followed by an increase at 4 °C min^−1^ to 240 °C.

Fatty acids were identified primarily by comparison of retention times with authentic FAME standards, and only well-resolved peaks consistently detected were reported. Fatty acid concentrations were expressed as mg g^−1^ dry weight, and percentage composition was calculated as a proportion of total identified fatty acids. Due to limited biomass, samples were pooled prior to analysis; therefore, fatty acid data are presented descriptively without statistical comparison between stocking densities.

Unsaturated fatty acids (UFA) were defined as the sum of monounsaturated fatty acids (MUFA) and polyunsaturated fatty acids (PUFA). Trans-fatty acids were excluded from interpretation because their detection was considered likely to result from analytical artifacts or contamination rather than biological origin in fairy shrimp.

To verify fatty acid identity, representative samples were independently analyzed at an ISO/IEC 17025–accredited laboratory (Central Laboratory (Thailand)) following AOAC Official Method 996.06, which confirmed the presence and relative abundance of major fatty acids detected in this study.

### 2.9. Carotenoid Content Analysis

Carotenoid content in *S. sirindhornae* was analyzed following the method described by Rodriguez-Amaya and Kimura [29], with minor modifications. Whole shrimp samples were homogenized and extracted with 10 mL of acetone under dim light to prevent pigment degradation. The extracts were filtered and analyzed using high-performance liquid chromatography (HPLC). The mobile phase consisted of acetonitrile, dichloromethane, and methanol, delivered at a flow rate of 1.0 mL min^−1^. Detection was performed at 450 nm using a photodiode array detector. Individual carotenoids were identified by comparing retention times and UV–vis absorption spectra with those of authentic standards. Results were expressed in µg g^−1^ dry weight.

### 2.10. Statistical Analysis

Growth-related variables (wet weight and total length) and water quality parameters were analyzed using one-way analysis of variance (ANOVA), followed by Duncan’s multiple range test to compare means among sampling days. Differences between stocking density treatments were assessed using independent-samples *t*-tests. Prior to conducting all parametric tests, data were examined for normality and homogeneity of variances using the Shapiro–Wilk test and Levene’s test, respectively.

All statistical analyses were performed using IBM SPSS Statistics version 28.0 (IBM Corp., Armonk, NY, USA).

To evaluate compositional differences between the gut content and pond plankton communities, multivariate analyses including hierarchical *k*-means clustering and principal component analysis (PCA) were conducted based on presence–absence matrices. These analyses were performed using RStudio version 3.6.1 (R Core Team, Vienna, Austria).

## 3. Results

### 3.1. Growth Performance of Fairy Shrimp

Significant differences in both body weight and total length were observed between fairy shrimp reared at low (20 ind. L^−1^) and high (40 ind. L^−1^) stocking densities (*p* < 0.05) (Figure 1). Individuals cultured at low density exhibited significantly higher body weight than those at high density on days 3 (*t*(58) = −2.242, *p* = 0.029), 9 (*t*(58) = −4.599, *p* < 0.001), 12 (*t*(58) = −4.411, *p* < 0.001), and 15 (*t*(58) = −4.792, *p* < 0.001). No significant difference in body weight was observed on day 6 (*p* > 0.05) (Figure 1a). The highest mean body weights were recorded on day 15, measuring 0.074 ± 0.013 g at low density and 0.059 ± 0.009 g at high density.

Similarly, total body length was significantly greater in shrimp reared at low density across all sampling days (*p* < 0.05) (Figure 1b). Maximum lengths were recorded on day 15, reaching 20.97 ± 0.90 mm at low density and 19.75 ± 0.61 mm at high density. Both groups exhibited continuous and significant increases in total length over the 15-day culture period (one-way ANOVA, *F*(5,174) = 79.009, *p* < 0.001).

### 3.2. Food Composition in the Gut Contents

A total of 54 taxa were identified from the gut contents of *S. sirindhornae* across both stocking densities. Specifically, 49 taxa were recorded in the low-density treatment and 48 taxa in the high-density group (Appendix A). The overall gut content composition is summarized in Table 1. Fairy shrimp under both density conditions exhibited similar dietary profiles, with phytoplankton as the predominant component, comprising 91.67–91.84% of the total gut contents, while zooplankton accounted for 8.16–8.33%.

Among the phytoplankton groups, chlorophytes were the most diverse and dominant (42.75–43.75%), followed by bacillariophytes (16.67–18.37%), euglenophytes (14.29–14.58%), cyanophytes (12.24–12.50%), chrysophytes (2.04–2.08%), and pyrophytes (2.04–2.08%). In contrast, zooplankton were represented primarily by rotifers (6.25–8.16%) and to a lesser extent by copepods (2.08%).

### 3.3. Food Diversity in the Cultural Pond

A total of 105 taxa of potential food organisms were identified in the culture ponds over six sampling intervals. Of these, 54 taxa were phytoplankton and 51 were zooplankton. The number of phytoplankton taxa ranged from 15 to 46 per sampling event. Chlorophytes were the most diverse group, comprising 27 taxa (50%), followed by bacillariophytes (9 taxa, 16.67%), cyanophytes (7 taxa, 12.96%), and charophytes (7 taxa, 12.96%). The most frequently encountered phytoplankton taxa included *Pediastrum* sp.1 (94.94% of samples), *Phacus* sp.1 (94.94%), *Closterium* sp. (88.89%), *Phacus* sp.2 (88.89%), and *Euglena* sp. (83.33%) (Appendix A).

Zooplankton diversity ranged from 27 to 38 taxa during the experimental period. Rotifers exhibited the greatest richness, with 33 species (63.46%), followed by cladocerans (8 species, 15.38%), copepods (7 taxa, 13.46%), protozoa (3 taxa, 5.77%), and ostracods (1 taxon, 1.92%). The most frequently observed zooplankton species were *Brachionus bidentatus* (100% of samples), followed by *Polyarthra vulgaris* (94.44%), *Moina micrura* (94.44%), and *Mesocyclops* sp. (88.89%) (Appendix A).

### 3.4. Food Compositions in the Fairy Shrimp Gut and the Cultural Pond

Principal component analysis (PCA) revealed significant differences in food compositions between the gut content and pond plankton communities across five sampling points (permutation test, *p* < 0.05) (Figure 2). Patterns of food occurrence in both low- and high-density treatments were generally consistent. In the high-density treatment, the first principal component (PC1) explained 40.28–50.33% of total variation, while the second component (PC2) accounted for 12.78–17.21%. Similarly, for the low-density group, PC1 and PC2 explained 34.55–53.79% and 12.91–16.22% of total variation, respectively.

Most food taxa—including phytoplankton and zooplankton—were associated with the pond samples; however, some key taxa were consistently identified in the gut contents. The similarity index between the gut and pond food composition fluctuated during the culture period, with the highest similarity (0.716) on day 3 and the lowest (0.569) on day 6. From day 9 onward, values stabilized between 0.636 and 0.667 (Table 2), suggesting a convergence between ingested food and environmental availability over time.

Notably, dominant phytoplankton taxa such as *Chlorella* sp., *Monoraphidium* sp., *Frustulia* sp., and *Navicula* sp. were consistently detected in both pond water and gut contents, highlighting their role as primary food sources. While PCA indicated divergence between gut and pond composition, the recurring presence of specific taxa in the gut implies possible selective ingestion. Nonetheless, such selectivity could also be influenced by differential digestion rates or capture efficiency. Therefore, while feeding preference is suggested, further controlled-choice experiments are needed to confirm active selectivity.

### 3.5. Biochemical Composition

The amino acid composition of *S. sirindhornae* is summarized in Table 3. Fairy shrimp fed on natural food in earthen ponds exhibited relatively low quantities of some amino acids, while the most abundant were glutamic acid, aspartic acid, lysine, and leucine, together accounting for 44.15% of the total amino acid content. Among the essential amino acids, high proportions of arginine, isoleucine, leucine, threonine, and valine were observed, although methionine and tryptophan were not detected, likely due to degradation during acid hydrolysis. The fatty acid profile of *Streptocephalus sirindhornae* (Table 4) was dominated by unsaturated fatty acids (90.15%), particularly monounsaturated fatty acids. Palmitoleic acid (C16:1n7) was the most abundant fatty acid, accounting for 56.26% of the total identified fatty acids. Linoleic acid (C18:2n6c), an omega-6 polyunsaturated fatty acid, represented 18.28% of the total profile and constituted the principal PUFA detected. Among saturated fatty acids, heptadecanoic acid (C17:0) was the dominant component (7.03%). These results are consistent with independent fatty acid analysis conducted by an ISO/IEC 17025–accredited laboratory using AOAC Official Method 996.06. Because biomass per replicate was insufficient, samples were pooled prior to analysis; therefore, fatty acid data are presented descriptively and are not intended for statistical comparison between stocking densities. In the case of carotenoid content, a higher level of total carotenoids was observed in the fairy shrimp fed natural food, measuring 155.91 µg g^−1^ dried weight. The primary carotenoid identified was β-carotene (83.24% of the total), whereas lutein and β-carotene were relatively low (Table 5).

### 3.6. Biomass Production

The biomass production of *S. sirindhornae* varied significantly between stocking densities. Fairy shrimp reared at 40 ind. L^−1^ achieved higher total biomass (610.47 ± 36.35 g) and yield (0.61 ± 0.03 kg m^−3^), reflecting greater production per unit volume. In contrast, shrimp cultured at 20 ind. L^−1^ exhibited significantly higher relative weight gain (403.11 ± 14.67%) despite lower biomass (0.41 ± 0.02 kg m^−3^), indicating superior individual growth performance under reduced density conditions (*p* < 0.05) (Table 6).

### 3.7. Water Quality Variables

Significant temporal variations were observed in the physicochemical properties of the pond water across the six sampling occasions (*p* < 0.05) (Figure 3). Water temperature increased steadily from 23.6–23.9 °C on day 3 to a peak of 29.8 °C on day 15 (*F*(5, 12) = 401.843, *p* < 0.001). The pH decreased from an initial value of 8.23 (day 0) to 7.60 on day 6, followed by a progressive increase thereafter (*F*(5, 12) = 100.180, *p* < 0.001). Dissolved oxygen declined significantly from 7.02 mg L^−1^ on day 0 to 3.83 mg L^−1^ on day 6 and dropped to near 0 mg L^−1^ from day 9 onward (*F*(5, 12) = 52.474, *p* < 0.001).

Electrical conductivity and total dissolved solids exhibited continuous increases throughout the experiment, reaching maxima of 1591.67 µS cm^−1^ and 796.33 mg L^−1^, respectively, on day 15 (*p* < 0.001). Salinity (≈1.0 ppt) and hardness (≈25 mg L^−1^) remained stable across all sampling times. Turbidity fluctuated between 15.2 and 28.3 NTU, peaking significantly on day 12 (*F*(5, 12) = 17.514, *p* < 0.001). Ammonia concentrations remained below 0.05 mg L^−1^, whereas nitrite was detected only on day 15 at 0.02 mg L^−1^.

## 4. Discussion

The fairy shrimp, *S. sirindhornae*, has been commercially produced as a live feed to replace brine shrimp in freshwater aquaculture [30]. The effects of different stocking densities on the growth performance of fairy shrimp were significant, particularly with respect to body weight and total body length. Fairy shrimp cultured at low stocking density (20 ind. L^−1^) exhibited significantly greater individual body weight than those at high density (40 ind. L^−1^) on days 3, 9, 12, and 15. These results align with those of Saejung et al. [31], who reported that shrimp fed with *Chlorella vulgaris* achieved the highest growth rate (0.94 ± 0.18 mm/day) on day 15. Similarly, Prompanya et al. [32] found that *Branchinella thailandensis* fed with *Chlorococcum humicola* at 2 × 10^6^ cells mL^−1^ over 15 days showed the best performance in growth, reproduction, survival, and nutritional content. Sriputhorn and Sanoamuang [8] also demonstrated improved growth and carotenoid accumulation in *M. rosenbergii* fed with fairy shrimp. In our study, shrimp at low density reached 0.074 g by day 15 compared to 0.059 g in the high-density group. Saengphan et al. [11] similarly reported that stocking density influenced *S. sirindhornae* performance when used as live feed for prawns.

Regarding food composition, 105 taxa were identified in the culture ponds, consisting of 54 phytoplankton and 51 zooplankton taxa, with chlorophytes being the most diverse group. In fairy shrimp guts, 54 taxa were detected, predominantly phytoplankton (>91%), reflecting pond composition. Prior studies [27,32,33] have also reported that fairy shrimp and *Branchinella* primarily consume phytoplankton, especially *Chlorella* sp., suggesting non-selective, omnivorous feeding habits. This supports the idea that environmental food availability directly shapes gut contents.

Our PCA showed distinct differences between pond and gut plankton compositions, especially on days 3 and 6. While similarity indices increased over time, the data do not provide sufficient evidence for strong feeding selectivity. Differences may result from digestion rates, ingestion mechanics, or sampling resolution. Controlled experiments are needed to clarify this behavior. The similarity in gut contents across treatments suggests that stocking density did not alter feeding behavior, though growth performance differed.

Biochemical analysis revealed high levels of amino acids—particularly glutamic acid, aspartic acid, lysine, and leucine—comparable to those reported in previous studies [9,12]. Lysine was especially abundant, supporting its nutritional value. Unsaturated fatty acids (notably omega-6) and carotenoids, especially β-carotene, were prominent, reflecting the availability of high-quality natural food. This is consistent with findings that *S. sirindhornae* can contribute to pigmentation in fish [12].

The relatively elevated proportions of odd-chain fatty acids, particularly C17:0 and C17:1, are uncommon but have been reported in aquatic organisms and are often associated with microbial contributions within natural food webs or dietary inputs in pond-based systems. In the present study, the presence of C17:0 and C17:1 was confirmed both by GC–MS analysis and by independent testing at an accredited laboratory, supporting their biological occurrence rather than analytical misidentification. Nevertheless, future studies employing additional chromatographic confirmation would further strengthen the resolution of structurally similar fatty acid isomers.

Water quality varied over the experimental period, with rising temperature and declining dissolved oxygen. However, all parameters remained within acceptable ranges for *S. sirindhornae* culture. Temperature increases were attributed to natural variation, and the temporary dip in DO was typical of small pond systems.

Biomass production was higher at 40 ind. L^−1^ (0.61 ± 0.03 kg m^−3^), though individual growth was greater at 20 ind. L^−1^. These findings are consistent with density-dependent growth limitations due to competition and oxygen availability [31,32,34]. Previous studies [35,36] on *Branchinella* and *Streptocephalus proboscideus* confirm that moderate crowding can optimize yield without harming survival. Thus, a stocking density of 30–40 ind. L^−1^ may offer a balance between yield and biological efficiency for semi-commercial production.

## 5. Conclusions

This study confirms the potential of *Streptocephalus sirindhornae* as a promising live feed organism for tropical freshwater aquaculture. Stocking density markedly influenced growth, yield, and biochemical composition. Low-density rearing (20 ind. L^−1^) improved individual growth and physiological condition, while high-density culture (40 ind. L^−1^) enhanced total biomass yield, suggesting a trade-off between growth efficiency and production output.

Nutritionally, *S. sirindhornae* demonstrated favorable profiles, high levels of essential amino acids, polyunsaturated fatty acids, and β-carotene, underscoring its suitability as a nutrient source for larval fish and crustaceans. Despite apparent discrepancies between pond plankton and gut contents, PCA results suggest that feeding patterns may be shaped by availability, ingestion capacity, or digestive selectivity. However, definitive conclusions on feeding preference require further controlled experimentation.

From an application standpoint, low-density culture combined with diverse phytoplankton availability—especially chlorophytes—can support efficient growth with minimal input, contributing to a sustainable aquaculture model. These findings provide a foundation for optimizing culture practices and enhancing the use of native fairy shrimp in live-feed production.

## Figures and Tables

**Figure 1 biology-15-00117-f001:**
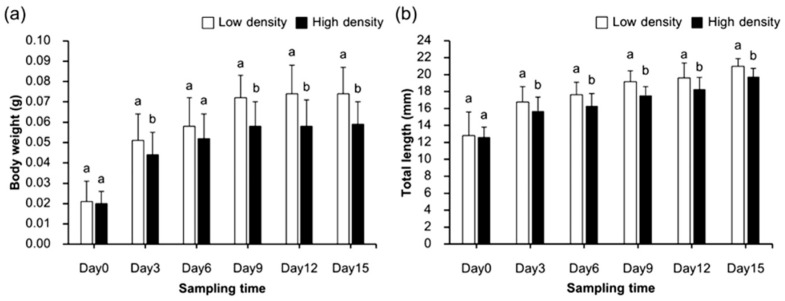
Comparison of (**a**) body weight per individual and (**b**) total body length of fairy shrimp *S. sirindhornae* under two different stocking densities. Different letters above the standard deviation bars indicate significant differences (*p* < 0.05).

**Figure 2 biology-15-00117-f002:**
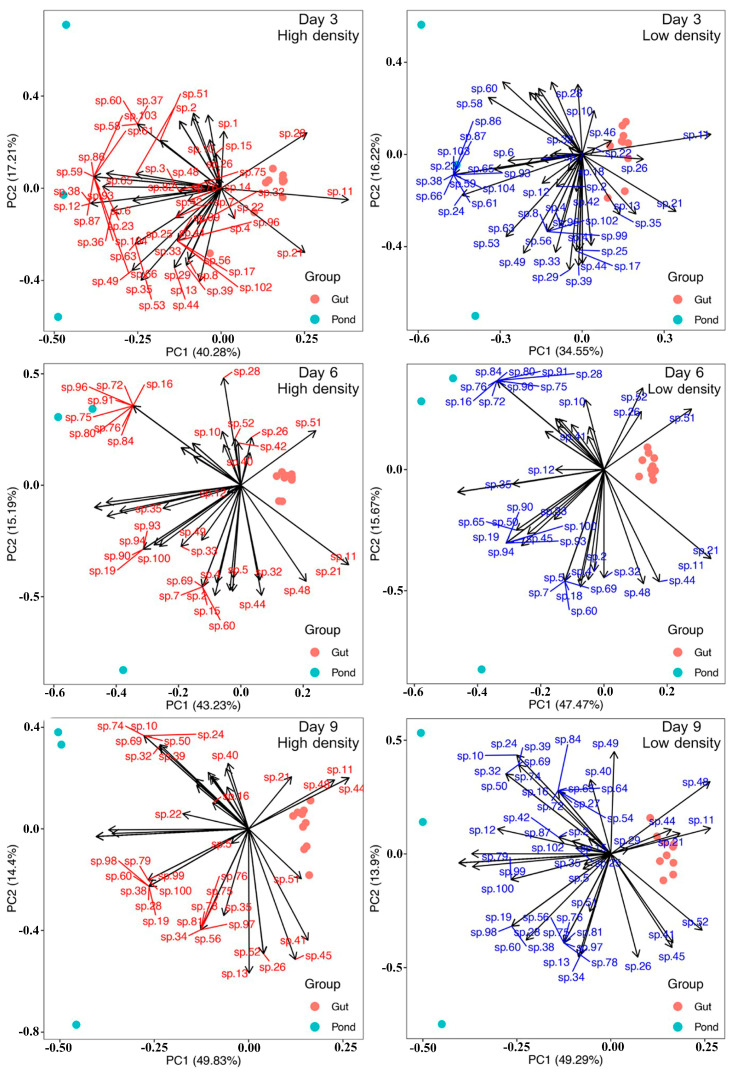
Principal component analysis (PCA) illustrating the relationships between plankton composition in the gut contents and in the pond. Taxa sp. 1–54 indicate phytoplankton taxa, and sp. 56–105 represent zooplankton taxa, as shown in Appendix A, respectively.

**Figure 3 biology-15-00117-f003:**
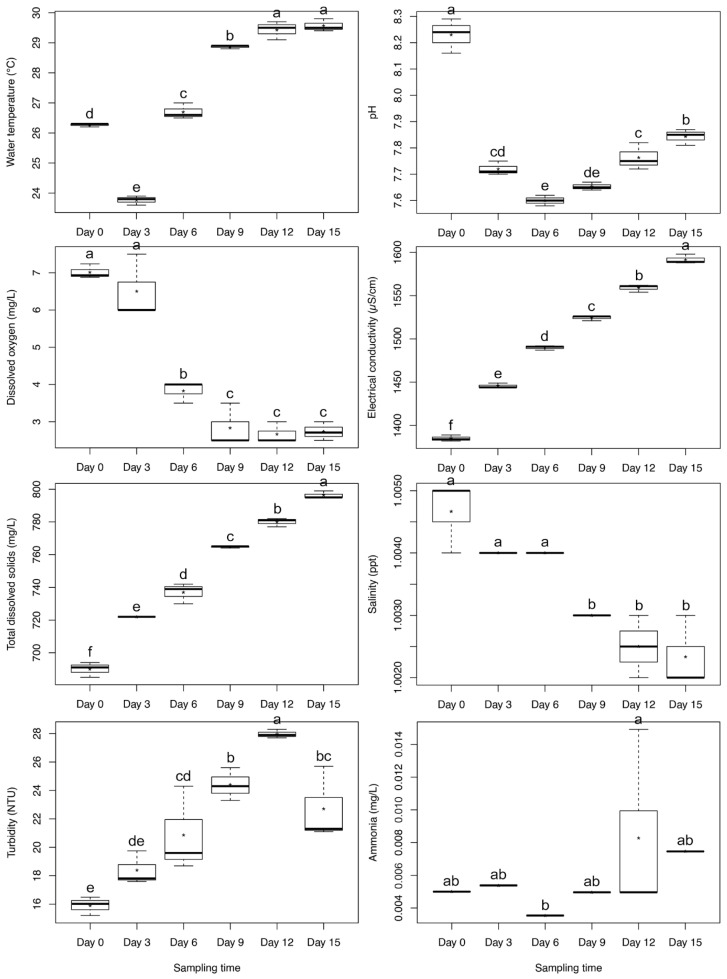
Boxplots showing variations in pond water physicochemical parameters across six sampling times. Different letters above boxplots indicate significant differences (*p* < 0.05). Asterisks (*) denote mean values.

**Table 1 biology-15-00117-t001:** The number of taxa and percentage composition of foods in the gut of *Streptocephalus sirindhornae* cultured at two different stocking densities.

Food Items	Low Density		High Density	
	Taxon Richness	%	Taxon Richness	%
Phytoplankton				
Cyanophyta	6	12.24	6	12.50
Chlorophyta	21	42.86	21	43.76
Euglenophyta	7	14.29	7	14.58
Chrysophyta	1	2.04	1	2.08
Bacillariophyta	9	18.37	8	16.67
Pyrrhophyta	1	2.04	1	2.08
Zooplankton				
Rotifera	4	8.16	3	6.25
Copepoda	–	–	1	2.08

**Table 2 biology-15-00117-t002:** Summary of the similarity between food items found in the guts and miscellaneous items collected from the earthen pond.

	Sampling Time				
	Day 3	Day 6	Day 9	Day 12	Day 15
Number of taxa co-occurrence	39	31	33	34	42
Similarity index	0.716	0.569	0.641	0.636	0.667

**Table 3 biology-15-00117-t003:** Amino acid composition (mg g^−1^ dry weight and %) of *S. sirindhornae* cultured in the earthen pond for 15 days. Essential amino acids (EAA) are indicated.

Amino Acid	*S. sirindhornae*		Essential Amino Acid (%)
	mg g^−1^	%	
Alanine	5.70	8.40	-
Arginine	4.27	6.29	12.97%
Aspartic acid	7.24	10.67	-
Glutamic acid	10.40	15.33	-
Glycine	3.53	5.20	-
Histidine	2.21	3.26	6.71%
Isoleucine	3.14	4.62	9.53%
Leucine	6.09	8.97	18.49%
Lysine	6.23	9.18	18.91%
Phenylalanine	3.54	5.22	10.75%
Proline	2.53	3.73	-
Serine	2.80	4.12	-
Threonine	3.35	4.93	10.16%
Tyrosine	2.72	4.01	-
Valine	4.11	6.05	12.48%

**Table 4 biology-15-00117-t004:** Concentrations (mg g^−1^ dry weight) and percentage composition (%) of fatty acids in the experimented *S. sirindhornae* (cultured in the earthen pond for 15 days).

Fatty Acid Composition	*S. sirindhornae*	
	mg g^−1^	%
Lauric acid (C12:0)	0.10	1.41
Arachidic acid (C20:0)	0.10	1.41
Heptadecanoic acid (C17:0)	0.50	7.03
Myristoleic acid (C14:1)	0.60	8.44
Pentadecanoic acid (C15:1n10)	0.21	2.95
Heptadecenoic acid (C17:1n10)	0.30	4.22
Palmitoleic acid (C16:1n7)	4.00	56.26
Linoleic acid (C18:2n6c)	1.30	18.28
Σ Saturated fatty acids (SFA)	0.70	9.85
Σ Monounsaturated fatty acids (MUFA)	5.11	71.87
Σ Polyunsaturated fatty acids (PUFA)	1.30	18.28
Σ Unsaturated fatty acids (UFA)	6.41	90.15
**Total identified fatty acids**	**7.11**	**100.00**

**Note:** Percentages were calculated as % of total identified fatty acids. UFA = MUFA + PUFA. Trans-fatty acids were excluded from interpretation.

**Table 5 biology-15-00117-t005:** Total carotenoid contents (μg g^−1^ dry weight) and percentage composition (%) of carotenoid profiles in the experimented *S. sirindhornae* (cultured in the earthen pond for 15 days).

Carotenoid Composition	*S. sirindhornae*
Lutein (%)	6.41
*β*-carotene (%)	83.24
Astaxanthin (%)	10.34
Total carotenoid contents (μg g^−1^ dry weight)	155.91

**Table 6 biology-15-00117-t006:** Growth performance of *Streptocephalus sirindhornae* cultured under different stocking densities.

Parameter	40 ind L^−1^	20 ind L^−1^	Significance (*p*)
Initial total weight (g)	205.00 ± 0.00	102.50 ± 0.00	—
Final total weight (g)	815.47 ± 36.35	515.68 ± 15.04	—
Weight gain rate (WGR, g)	610.47 ± 36.35	413.18 ± 15.04	—
% Weight gain rate (WGR%)	297.79 ± 17.73	403.11 ± 14.67 *	*p* < 0.05
**Total yield (kg m^−3^)**	**0.61 ± 0.03** *	**0.41 ± 0.02**	*p* < 0.05

**Note:** Values are expressed as mean ± SD. Asterisks (*) denote significant differences between treatments (*p* < 0.05). Total yield (kg m^−3^) = (Final total biomass − Initial biomass)/Volume of culture system (1 m^3^).

## Data Availability

Data supporting the findings of this study are available from the corresponding author upon reasonable request.

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
