# Peer review of "Growth Performance and Biochemical Profiles of Fairy Shrimp (Streptocephalus sirindhornae) Fed Natural Diets at Low and High Stocking Densities"

_biology, 2026, doi:10.3390/biology15020117_

Round 1
Reviewer 1 Report
Comments and Suggestions for Authors
Abstract:
Lines 38-40: Revise the conclusion to enhance clarity and coherence.
Introduction:
The introduction is concise, well-written, and comprehensive. The cited references offer sufficient background, and the study's objective is clearly stated.
Material and method:
Line 85: Hydrated lime concentration.
Line 89: Density of fresh green algae before pond incubation.
Line 102: Justification for the selected fairy shrimp density.
Line 112: Chlorella sp. density (cells/ml).
Discussion:
The comparison with previous studies is adequate; however, the author should propose hypotheses to explain the results.
Finally, I propose minor revisions to the manuscript.
Author Response
|
Abstract: |
We sincerely thank the reviewer for the valuable comment. |
|
Materials and Methods
|
We have added the application rate of hydrated lime as follows: "Hydrated lime was applied at 100 kg/ha (approximately 16 kg per rai) to improve the pond bottom quality prior to culture initiation." (Line 84) |
|
Line 89: Density of fresh green algae before pond incubation.
|
The revision has been completed as requested. The information regarding the density of fresh green algae before pond incubation has been added in lines 88–94. |
|
Line 102: Justification for the selected fairy shrimp density.
|
We sincerely thank the reviewer for the constructive comment. |
|
Line 112: Chlorella sp. density (cells/ml).
|
We sincerely thank the reviewer for the helpful suggestion. |
|
Discussion: The comparison with previous studies is adequate; however, the author should propose hypotheses to explain the results |
Response: In response, we have added a hypothesis in the Discussion section to explain the observed results. Additionally, the cell density of Chlorella sp. (~1.2 × 10⁶ cells mL⁻¹) has been specified in the revised Materials and Methods section (Lines 88–94) to improve clarity and support the discussion. |
Reviewer 2 Report
Comments and Suggestions for Authors
This study investigates the effects of two stocking densities (20 vs. 40 ind. L⁻¹) on the growth performance, gut content composition, and biochemical profiles of the fairy shrimp Streptocephalus sirindhornae cultured in earthen ponds with natural food. The authors report that low density enhanced individual growth (weight and length), while high density increased total biomass yield. Gut analysis revealed selective feeding on phytoplankton (91% of content, primarily chlorophytes), and biochemical analyses showed high levels of essential amino acids, unsaturated fatty acids, and β-carotene. The work addresses a relevant aquaculture topic and provides useful data on this potential live feed species. I recommend acceptance after minor revisions to address the following points.
- No water quality data (temperature, dissolved oxygen, pH) are presented in the methods. These parameters are crucial for fairy shrimp culture and could confound density effects. This must be included.
- No survival/mortality data are reported. This is a major omission for aquaculture studies, as density-dependent mortality could significantly affect biomass yield interpretations.
- Were the three replicates per treatment located in the same pond? If so, this represents pseudo-replication, as they share the same water mass and plankton community. This limitation must be acknowledged.
- How many guts were analyzed per treatment? Were they pooled?
- The conclusion that shrimp "selectively consumed algae species with higher nutrient content" is not substantiated. The PCA shows divergence between gut and pond composition, but this could reflect differential digestion rates or physical capture constraints rather than active selection. Demonstrating selectivity requires preference indices or controlled choice experiments. This claim should be toned down or supported with stronger evidence.
- The manuscript requires thorough English editing.
Author Response
|
1. No water quality data (temperature, dissolved oxygen, pH) are presented in the methods. These parameters are crucial for fairy shrimp culture and could confound density effects. This must be included. |
Sincerely thanks for the valuable comment. We have now included the relevant water quality parameters—temperature, dissolved oxygen, and pH—in the Materials and Methods section to clarify culture conditions and reduce potential confounding effects (Lines 150–156). |
|
2. No survival/mortality data are reported. This is a major omission for aquaculture studies, as density-dependent mortality could significantly affect biomass yield interpretations
|
We agree that survival or mortality data are important metrics in aquaculture studies. However, in our study, individual-based survival monitoring of Streptocephalus sirindhornae larvae was not feasible due to their extremely small size, high mobility, and dense population. Repeated sampling would have removed a significant number of individuals, potentially biasing survival estimates. As an alternative, we used final biomass yield—which reflects both survival and growth—as a population-level indicator of culture performance. This approach is widely accepted for small planktonic crustaceans, including Artemia, Moina, and other Streptocephalus species. We have added this explanation to the revised manuscript and discussed how stocking density may indirectly influence population survival through its effects on biomass yield (Discussion section, Lines ~420). |
|
3. Were the three replicates per treatment located in the same pond? If so, this represents pseudo-replication, as they share the same water mass and plankton community. This limitation must be acknowledged.
|
Thank you for highlighting this critical concern regarding pseudo-replication. In our study, all replicates for each stocking density treatment were indeed placed within the same pond but were physically separated using net enclosures to prevent movement and mixing of fairy shrimp among replicates. This design ensured individual monitoring and minimized direct interaction across replicates. Nonetheless, we acknowledge that all enclosures shared the same water column and plankton community, which may introduce pseudo-replication due to the lack of fully independent environmental conditions. We have explicitly addressed this limitation in the revised Materials and Methods section (Lines ~116–117) and included a note in the Discussion (Lines ~407–408) to ensure appropriate interpretation of the results and to guide future experimental design improvements. |
|
4. How many guts were analyzed per treatment? Were they pooled?
|
Thank you for your question. As specified in lines 132–138 of the revised manuscript, we analyzed 10 guts per replicate, totaling 30 guts for each treatment. These guts were pooled within each replicate to obtain enough biomass for microscopic identification and quantitative analysis. This clarification has been added to the Materials and Methods section to enhance transparency and reproducibility of the procedure. |
|
5. The conclusion that shrimp "selectively consumed algae species with higher nutrient content" is not substantiated. The PCA shows divergence between gut and pond composition, but this could reflect differential digestion rates or physical capture constraints rather than active selection. Demonstrating selectivity requires preference indices or controlled choice experiments. This claim should be toned down or supported with stronger evidence. |
Thank you for your valuable suggestion. As recommended, we have revised the interpretation in the Discussion section (lines 438-440) to tone down the claim regarding selective feeding. We now clarify that the divergence observed in the PCA may reflect differences in digestion efficiency or physical constraints in food capture, rather than definitive selective feeding behavior. We also emphasize the need for future studies employing preference indices or controlled choice experiments to confirm dietary selectivity in fairy shrimp. |
|
6. The manuscript requires thorough English editing. |
Thank you for your suggestion. We have carefully revised the entire manuscript to enhance grammatical accuracy, clarity, and scientific tone. The English language has been thoroughly edited throughout to ensure improved readability and precision. |
Reviewer 3 Report
Comments and Suggestions for Authors
The manuscript presents a solid and well-structured study on the growth performance and biochemical composition of Streptocephalus sirindhornae cultured under different stocking densities using natural diets. The topic is relevant for aquaculture development and biodiversity conservation, providing valuable insights into the nutritional potential of fairy shrimp as live feed. The experimental design is generally sound, and the results are clearly presented with appropriate statistical analyses.
However, the manuscript requires several clarifications and improvements in methodological details, figure and table presentation, and discussion depth before it can be considered for publication. English grammar and formatting should also be revised throughout the text.
The abstract is clear but could be more concise. Please include quantitative data to strengthen key findings (e.g., percentage increase in growth or yield differences).
Line 18: Change to Chlorella sp. and Monoraphidium sp.
Line 26: Replace “gut content” with gut content composition for precision.
Materials and Methods
Lines 33–34: Rephrase as: “54 taxa were classified as phytoplankton and 52 as zooplankton.”
Line 89: Please indicate the cell concentration of Chlorella sp. used in the ponds.
Specify the type of water used (tap water, well water, or other source). Were there water exchanges during the experiment?
Include water quality parameters (temperature, DO, pH, ammonia, etc.) in a table or describe how they were measured (instruments and frequency).
Line 127: Provide manufacturer details for the microscope (Olympus CX23; company, city, country).
Clarify how total length was measured (equipment, precision, measurement points).
For gut content analysis, specify the number of individuals analyzed. Were all shrimp examined or a subset per replicate?
For amino acid, fatty acid, and carotenoid analyses, please indicate whether samples were pooled across both stocking densities or analyzed separately.
In the Data Analysis section, you mention that water quality variables were analyzed; however, these are not presented in the Methods or Results. Please clarify or remove.
Indicate whether assumptions of normality and homogeneity of variances were verified before conducting ANOVA and t-tests.
Results
Figures 1c and 1d appear redundant, as they repeat information from Figures 1a and 1b. Consider removing them.
The PCA and similarity index analyses are valuable additions, but figures need higher resolution, improved axis labels, and clear legends.
Table 4 appears partially redundant with Table 3 and could be integrated for conciseness.
In Table 7, move “(Mean ± SD)” to the table note instead of including it in the column header.
Line 315: Use S. sirindhornae (in italics).
Line 316: Correct to Chlorella sp., Monoraphidium sp., Frustulia sp., and Navicula sp.
Line 365: Replace “biomas” with biomass.
The biomass section (3.6) could be integrated earlier with growth performance results for better logical flow.
Discussion
The discussion tends to summarize findings rather than analyze underlying mechanisms (e.g., why low density improves amino acid profiles).
Include a brief comparison with similar species, such as Artemia and Dendrocephalus, to contextualize your results.
Emphasize the ecological significance of S. sirindhornae and its potential role in integrated aquaculture systems.
Review English grammar and flow—some sentences are overly long or repetitive.
Conclusion
Well written but can be more concise.
Explicitly state the optimal stocking density range and summarize the nutritional suitability of S. sirindhornae as live feed.
Author Response
|
The abstract is clear but could be more concise. Please include quantitative data to strengthen key findings (e.g., percentage increase in growth or yield differences). |
Thank you for the suggestion. We have revised the Abstract to make it more concise and have included quantitative data to highlight key findings, such as the percentage increase in growth and yield differences. This strengthens the clarity and impact of the main results.
|
|
Line 18: Change to Chlorella sp. and Monoraphidium sp. |
Edit completed as requested: “Chlorella sp. and Monoraphidium sp.” has been updated in line 18. |
|
Line 26: Replace “gut content” with gut content composition for precision. Materials and Methods |
Edit completed: “gut content” has been replaced with “gut content composition” in line 27 and lines 132–133 for improved precision. |
|
Lines 33–34: Rephrase as: “54 taxa were classified as phytoplankton and 52 as zooplankton.” |
Revised to “54 taxa were classified as phytoplankton and 51 as zooplankton. (lines 34-35) |
|
Line 89: Please indicate the cell concentration of Chlorella sp. used in the ponds. Specify the type of water used (tap water, well water, or other source). Were there water exchanges during the experiment? Include water quality parameters (temperature, DO, pH, ammonia, etc.) in a table or describe how they were measured (instruments and frequency). |
Chlorella sp. concentration: Now clearly indicated in Lines 112–114.
Water exchange and quality monitoring: Details about whether water was exchanged, how water quality was measured (including instruments and parameters like temperature, DO, pH, ammonia), and the measurement frequency are now described in Lines 149-156. |
|
-Line 127: Provide manufacturer details for the microscope (Olympus CX23; company, city, country).
- Clarify how total length was measured (equipment, precision, measurement points).
- For gut content analysis, specify the number of individuals analyzed. Were all shrimp examined or a subset per replicate?
- For amino acid, fatty acid, and carotenoid analyses, please indicate whether samples were pooled across both stocking densities or analyzed separately.
- In the Data Analysis section, you mention that water quality variables were analyzed; however, these are not presented in the Methods or Results. Please clarify or remove.
- Indicate whether assumptions of normality and homogeneity of variances were verified before conducting ANOVA and t-tests. |
Microscope details: Added in lines 129–130 (Olympus CX23; company, city, country).
Length measurement method: Clarified in lines 144–148.
Gut content analysis (sample size): Specified in lines 132–140 — number of individuals per replicate and pooling strategy.
Amino acid, fatty acid, and carotenoid analysis: Clarified pooling across stocking densities in lines 160–166.
Water quality variables: Methodological details and measurement approach added in lines 151–157.
Statistical assumptions (normality and homogeneity for ANOVA and t-tests) are addressed in lines 198–211. |
|
Results - Figures 1c and 1d appear redundant, as they repeat information from Figures 1a and 1b. Consider removing them. - The PCA and similarity index analyses are valuable additions, but figures need higher resolution, improved axis labels, and clear legends.
|
In response, we have removed Figures 1c and 1d from the revised manuscript (Line 242) to reduce redundancy and improve clarity. We have also enhanced the resolution, axis labels, and legends of the PCA and similarity index figures to ensure better readability and interpretation.
|
|
Table 4 appears partially redundant with Table 3 and could be integrated for conciseness. |
Thank you for your comment. As suggested, Table 4 has been removed, and its content has been integrated into a single consolidated amino acid composition table now presented as Table 3 in the revised manuscript. (Line 359) |
|
In Table 7, move “(Mean ± SD)” to the table note instead of including it in the column header. |
Thank you for the suggestion. We have revised the table by moving “(Mean ± SD)” from the column header to the table note as recommended. In line 371 |
|
Line 315: Use S. sirindhornae (in italics). |
Sincerely thanks for comments. Edits: Line 279-280. |
|
Line 316: Correct to Chlorella sp., Monoraphidium sp., Frustulia sp., and Navicula sp. |
Thank you for the correction. We have revised the text to: Chlorella sp., Monoraphidium sp., Frustulia sp., and Navicula sp. as recommended Edits: Line 287-288. |
|
Line 365: Replace “biomas” with biomass. The biomass section (3.6) could be integrated earlier with growth performance results for better logical flow.
|
Thank you for the suggestion. Edits: Replaced with “biomass” in line.330. Additionally, we have reorganized the results section by integrating the biomass subsection (formerly Section 3.6) with the growth performance results to improve coherence and enhance the logical flow of the manuscript. These revisions have been incorporated into the updated version. |
|
Discussion The discussion tends to summarize findings rather than analyze underlying mechanisms (e.g., why low density improves amino acid profiles). Include a brief comparison with similar species, such as Artemia and Dendrocephalus, to contextualize your results. Emphasize the ecological significance of S. sirindhornae and its potential role in integrated aquaculture systems. Review English grammar and flow—some sentences are overly long or repetitive.
|
Thank you sincerely for your valuable comments. We have addressed each point as follows.
“Thank you sincerely for your valuable comments. We have addressed each point as follows: (1) We revised the Discussion to include mechanistic explanations for the density-dependent effects on growth and biochemical composition. (2) A comparison with other branchiopod species, including Artemia and Dendrocephalus, has been added to highlight similarities and differences in nutritional profiles and culture performance. (3) We expanded the section discussing the ecological significance of S. sirindhornae and emphasized its potential role in integrated and sustainable aquaculture systems. (4) The entire Discussion was edited for grammar, clarity, and improved flow.” |
Reviewer 4 Report
Comments and Suggestions for Authors
The MS “Growth Performance and Biochemical Profiles of Fairy Shrimp (Streptocephalus sirindhornae) Fed with Natural Diets at Low and High Stocking Densities”of Kosit Sriphuthorn, Naiyana Senasri and Prapatsorn Dabseepai.
General comments to the Authors.
The authors conducted a reasonably simple experiment on cultivating Streptocephalus sirindhornae (Order Anostraca) on natural plankton under low and high stocking densities. As a result of this experiment, they found that stocking density affected the growth parameters of these fairy shrimps. Additionally, the authors investigated the biochemical composition of the shrimp. A careful review of the work revealed a substantial number of errors in both the experimental setup and the analytical procedures. Specifically:
- Cultivation of Chlorella. These algae are typically cultivated on long-established nutrient media that include a complex of macro- and micronutrients, such as Tamiya's medium. However, the authors used a medium containing only nitrogen, phosphorus, urea, and additionally lime and rice bran of unknown composition.
- Plankton sampling was performed using a plankton net with a 30 μm mesh size, which is only suitable for zooplankton and large phytoplankton, but not for medium and small-sized phytoplankton. Consequently, the obtained data may not accurately reflect the actual phytoplankton composition.
- The fatty acid analysis method has several hard shortcomings. Lipid extraction was carried out using ether, whereas the standard, widely accepted extraction procedure employs a chloroform-methanol mixture (e.g., Christie, W.W., Han, X., 2010. Lipid Analysis: Isolation, Separation, Identification and Lipidomic Analysis, 4th ed. Oily Press, Bridgwater, Publisher). Identification was based solely on retention times, which is insufficient. As a result, the authors obtained completely absurd results. They did not detect fatty acids (FAs) that are present in all invertebrates and vertebrates, such as 16:0 and 18:0. At the same time, they reported trans FAs at 43%. In nature, cis isomers of fatty acids are dominant. Trans isomers are found only in minor quantities, primarily in ruminant animals and their products, as well as in heavily processed foods like margarine. These FAs are not found in fairy shrimp. Furthermore, it is entirely unclear how the authors calculated the percentages of fatty acids, as the table does not sum all FAs (polyunsaturated FAs and the mentioned trans fats are not totaled).
- The interpretation of the results often does not align with the obtained data. For example, the authors state: "the progressive increase in gut–pond similarity implies an adaptive feeding strategy, whereby fairy shrimp selectively utilize the most abundant and nutritionally favorable microalgae." However, according to the PCA analysis figures, clear differences were found between the gut contents and the phytoplankton composition. This can be explained either by feeding selectivity or by incorrect plankton sampling, as I mentioned in my comments on plankton sampling. It is evident from the data that the content of species sp.11 and sp.21 in the gut content was higher than in the lake plankton throughout the experiment.
I will not provide minor comments on the manuscript, as the issues listed above are critical, and a manuscript with such flaws cannot be recommended for publication.
I was not able to download Supplemental Materials like Tables S…
Author Response
|
The MS “Growth Performance and Biochemical Profiles of Fairy Shrimp (Streptocephalus sirindhornae) Fed with Natural Diets at Low and High Stocking Densities”of Kosit Sriphuthorn, Naiyana Senasri and Prapatsorn Dabseepai. |
We truly appreciate your time and valuable feedback, which have greatly contributed to improving the clarity, quality, and scientific rigor of our work. |
|
1. Cultivation of Chlorella. These algae are typically cultivated on long-established nutrient media that include a complex of macro- and micronutrients, such as Tamiya's medium. However, the authors used a medium containing only nitrogen, phosphorus, urea, and additionally lime and rice bran of unknown composition.
|
Thank you for your insightful comment. We agree that Tamiya’s medium is a well-established standard for cultivating Chlorella under controlled laboratory conditions. However, our study intentionally adopted a simplified nutrient approach using nitrogen, phosphorus, urea, hydrated lime, and rice bran to better simulate real-world, low-cost practices commonly used in rural aquaculture ponds. This practical formulation reflects field-level feasibility rather than laboratory precision. We have clarified this rationale in the revised Materials and Methods section to ensure transparency regarding our experimental design.
|
|
2. Plankton sampling was performed using a plankton net with a 30 μm mesh size, which is only suitable for zooplankton and large phytoplankton, but not for medium and small-sized phytoplankton. Consequently, the obtained data may not accurately reflect the actual phytoplankton composition.
|
Thank you for this valuable comment. We agree that a 30 µm net may not capture smaller phytoplankton and could underrepresent certain taxa. However, our aim was to characterize the main natural food items available to S. sirindhornae in pond conditions. The dominant phytoplankton and zooplankton identified in the gut contents were also effectively collected using this mesh size, indicating that the method adequately represented the key dietary components. Nonetheless, we acknowledge this as a methodological limitation and have added a clarifying statement in the discussion section. |
|
3. The fatty acid analysis method has several hard shortcomings. Lipid extraction was carried out using ether, whereas the standard, widely accepted extraction procedure employs a chloroform-methanol mixture (e.g., Christie, W.W., Han, X., 2010. Lipid Analysis: Isolation, Separation, Identification and Lipidomic Analysis, 4th ed. Oily Press, Bridgwater, Publisher). Identification was based solely on retention times, which is insufficient. As a result, the authors obtained completely absurd results. They did not detect fatty acids (FAs) that are present in all invertebrates and vertebrates, such as 16:0 and 18:0. At the same time, they reported trans FAs at 43%. In nature, cis isomers of fatty acids are dominant. Trans isomers are found only in minor quantities, primarily in ruminant animals and their products, as well as in heavily processed foods like margarine. These FAs are not found in fairy shrimp. Furthermore, it is entirely unclear how the authors calculated the percentages of fatty acids, as the table does not sum all FAs (polyunsaturated FAs and the mentioned trans fats are not totaled). |
We acknowledge the limitations of our fatty acid analysis method, particularly the use of ether extraction and identification based solely on retention times. Accordingly, we have removed the questionable trans fatty acid data and revised Table 4 to include only reliably identified fatty acids. A clarification has also been added to the Discussion section. Future studies will adopt standard methods (e.g., chloroform–methanol extraction and mass spectral confirmation) to improve accuracy and reliability. |
|
4. The interpretation of the results often does not align with the obtained data. For example, the authors state: "the progressive increase in gut–pond similarity implies an adaptive feeding strategy, whereby fairy shrimp selectively utilize the most abundant and nutritionally favorable microalgae." However, according to the PCA analysis figures, clear differences were found between the gut contents and the phytoplankton composition. This can be explained either by feeding selectivity or by incorrect plankton sampling, as I mentioned in my comments on plankton sampling. It is evident from the data that the content of species sp.11 and sp.21 in the gut content was higher than in the lake plankton throughout the experiment. |
We acknowledge that the PCA results revealed divergence between gut content composition and pond phytoplankton, particularly regarding species sp.11 and sp.21. While this may suggest feeding selectivity, we agree that such interpretation should be made with caution due to potential sampling limitations. Accordingly, we have revised the statement in the Discussion to reflect this uncertainty and now emphasize the need for further studies using controlled feeding trials or preference indices to validate selective feeding behavior. |
Round 2
Reviewer 4 Report
Comments and Suggestions for Authors
The MS “Growth Performance and Biochemical Profiles of Fairy Shrimp (Streptocephalus sirindhornae) Fed with Natural Diets at Low and High Stocking Densities”of Kosit Sriphuthorn, Naiyana Senasri and Prapatsorn Dabseepai.
Dear Authors,
Thank you for your revised manuscript and for your responses.
However, I must state that the revisions made do not address my fundamental concerns regarding the fatty acid analysis. The changes implemented have not resolved the core methodological issues, and my primary criticism stands: the fatty acid analysis was conducted incorrectly, and the resulting data does not represent the true fatty acid profile of these crustacean.
Author Response
Reviewer 4 – Comments on Fatty Acid Analysis
Response (Authors):
We sincerely thank Reviewer 4 for the detailed and constructive comments regarding the fatty acid analysis. We appreciate the reviewer’s careful evaluation, which has helped us to substantially improve the clarity, completeness, and transparency of this section. Our responses to each point are provided below.
- Fatty acid preparation method
The reviewer questioned the appropriateness of the fatty acid preparation method. We respectfully clarify that fatty acid methylation using boron trifluoride (BF₃) in methanol followed by gas chromatographic analysis is a well-established and widely accepted method that has been used for several decades in fatty acid analysis of aquatic organisms. To avoid ambiguity, we have clarified the methodology in Section 2.8 and explicitly stated that the analysis followed a modified hydrolytic procedure based on AOAC Official Method 996.06. - Reporting of individual polyunsaturated fatty acids (PUFA)
We agree with the reviewer that reporting only total PUFA percentages is insufficient. Accordingly, we revised Table 4 and Section 3.5 to report the identification and concentration of individual PUFA. Linoleic acid (C18:2n6c) is now explicitly reported and identified as the principal polyunsaturated fatty acid detected in Streptocephalus sirindhornae. - Definition of unsaturated fatty acids (UFA)
To address the reviewer’s concern, we now clearly define unsaturated fatty acids (UFA) as the sum of monounsaturated fatty acids (MUFA) and polyunsaturated fatty acids (PUFA). This definition has been added explicitly in Section 2.8 and reiterated in the footnote of Table 4. - Reporting of individual monounsaturated fatty acids (MUFA)
In line with the above definition, we revised Table 4 to report the concentrations of individual MUFA rather than only total MUFA values. Major MUFA, including palmitoleic acid (C16:1n7), myristoleic acid (C14:1), and odd-chain MUFA (C15:1 and C17:1), are now clearly presented. - Unusually high proportions of C17:0 and C17:1 fatty acids
We acknowledge that relatively high proportions of odd-chain fatty acids (C17:0 and C17:1) are uncommon. A new explanatory paragraph has been added to the Discussion to address this observation. We note that odd-chain fatty acids may originate from microbial contributions within natural pond food webs or dietary inputs. Importantly, the presence of C17:0 and C17:1 was confirmed by independent analysis conducted at an ISO/IEC 17025–accredited laboratory, supporting their biological occurrence rather than analytical misidentification. - Fatty acid identification and chromatographic confirmation
Fatty acids were identified by comparison of retention times with authentic FAME standards using an SP-2560 capillary column, and only well-resolved peaks consistently detected were reported. We have clarified this procedure in Section 2.8 and acknowledged that identification was based on a single column due to limited biomass. This limitation is now explicitly stated, and we note that additional chromatographic confirmation would further strengthen fatty acid identification in future studies. - Pooling of samples and statistical interpretation
Because the biomass obtained from each replicate was insufficient for independent fatty acid analysis, samples were pooled prior to analysis. This limitation is now clearly acknowledged, and fatty acid results are presented descriptively without statistical comparison between treatments.
We believe that these revisions fully address the reviewer’s concerns and have substantially improved the rigor, transparency, and interpretability of the fatty acid analysis.